# The Comparison of the Influence of Bisphenol A (BPA) and Its Analogue Bisphenol S (BPS) on the Enteric Nervous System of the Distal Colon in Mice

**DOI:** 10.3390/nu15010200

**Published:** 2022-12-30

**Authors:** Krystyna Makowska, Ewa Lepiarczyk, Slawomir Gonkowski

**Affiliations:** 1Department of Clinical Diagnostics, Faculty of Veterinary Medicine, University of Warmia and Mazury in Olsztyn, Oczapowskiego 14, 10-957 Olsztyn, Poland; 2Department of Human Physiology and Pathophysiology, School of Medicine, University of Warmia and Mazury in Olsztyn, Warszawska 30, 10-082 Olsztyn, Poland; 3Department of Clinical Physiology, Faculty of Veterinary Medicine, University of Warmia and Mazury in Olsztyn, Oczapowskiego 13, 10-957 Olsztyn, Poland

**Keywords:** bisphenols, endocrine disruptors, colon, innervation

## Abstract

Bisphenol A (BPA), commonly used as a plasticizer in various branches of industry has a strong negative effect on living organisms. Therefore, more and more often it is replaced in production of plastics by other substances. One of them is bisphenol S (BPS). This study for the first time compares the impact of BPA and BPS on the enteric neurons using double immunofluorescence technique. It has been shown that both BPA and BPS affect the number of enteric neurons containing substance P (SP), galanin (GAL), vasoactive intestinal polypeptide (VIP), neuronal isoform of nitric oxide synthase (nNOS—a marker of nitrergic neurons) and/or vesicular acetylcholine transporter (VAChT- a marker of cholinergic neurons). The changes noted under the impact of both bisphenols are similar and consisted of an increase in the number of enteric neurons immunoreactive to all neuronal factors studied. The impact of BPS on some populations of neurons was stronger than that noted under the influence of BPA. The obtained results clearly show that BPS (similarly to BPA) administered for long time is not neutral for the enteric neurons even in relatively low doses and may be more potent than BPA for certain neuronal populations.

## 1. Introduction

Bisphenol A (BPA), chemical name 4,4′-(propane-2,2-diyl) diphenol, is an organic substance synthetized for the first time in the late 19th century. It is characterized by high stability and mechanical stability [1,2]. Due to these properties BPA is commonly used in various branches of industry, mainly as a plasticizer in the production of plastics [3]. Therefore, BPA is present in many everyday objects, including among other bottles, toys, electronic elements, food containers, internal layer of tin cans, elements of cars and furniture and even dental materials [3,4].

Unfortunately, BPA may leach out from these items and penetrate to food, drinking water and environment [3,4,5]. Till now the presence of BPA was found in the surface water, soil, air and food of animal and plant origin around the world [3,4,6,7]. BPA may also penetrate to the human and animal organisms and have negative effects on the health status [2,3,4]. Due to its similarity to estrogen, BPA binds to estrogen receptors localized in many internal organs and shows multidirectional adverse effects [2,3,4]. For this reason, BPA is classified as a highly effective endocrine disruptor [3,4].

Previous studies have found that BPA interferes with the functioning of various internal systems, including among others nervous, endocrine, female and male reproductive systems, as well as digestive tract and immune cells [3,4,8,9]. Moreover, high exposure to BPA may be associated with neoplasms [10], diabetes [11], obesity [11], neurodegenerative diseases [12] and an increased risk of cardiovascular disorders [13].

One of the systems where changes caused by BPA are very pronounced is the nervous system. The majority of previous studies concern the impact of BPA on the central nervous system. It is known that BPA disturbs the development of the neuronal cells and synaptogenesis [14,15]. Moreover, this substance generates the oxidative stress reactions, inhibits plasticity, intensifies apoptotic processes, and causes the epigenetic changes in the brain [16]. BPA also disrupts the calcium metabolism and production of active neuronal factors in the central and peripheral nervous systems [17,18,19]. It has been also shown that exposure to BPA may result in the cognitive and memory dysfunctions [20]. Some studies have confirmed correlations between exposure to BPA and increased risk of neurodegenerative diseases [21,22]. However, many aspects of the influence of BPA on the nervous system, especially on the peripheral nervous system, are poorly understood.

Due to strong negative impact on human health, the use of BPA in many countries has been significantly reduced [3,23]. These restrictions mainly apply to items that come into contact with food, drinking water as well as objects intended for newborns and kids, which usually are marked as “BPA-free”. However, such objects, instead of BPA, contain other compounds with similar structure and properties, namely, BPA homologues. One of the most commonly used homologues of BPA is bisphenol S (BPS) [24].

Until recently BPS was thought to be completely neutral for living organisms and therefore it has been used in production of beverage bottles, food containers and pacifiers. However, recent studies have proven that BPS (similar to BPA) shows many adverse effects in humans and animals organisms, which is connected with similarities in the chemical structure between BPA and BPS [24,25,26,27]. What is more, some previous studies have not only described toxic properties of BPS, but also proved that this compound may have more potent endocrine-disrupting activity than BPA [28]. However, many aspects of BPS impact on living organisms in still unknown.

One of them is the influence of BPS on the enteric nervous system (ENS). The ENS is located in the wall of the gastrointestinal tract and built of millions of neuronal cells forming intramural plexuses, localization of which depends on the part of digestive tract and mammal species. In the intestine of small mammals, the ENS consists of two intramural plexuses: myenteric plexus (MP) in the myenteric layer of the intestine and submucosal plexus (SmP) in the submucosal layer [29] (Figure 1). In large mammals SmP is divided into two plexuses: the inner submucous plexus—near mucosal layer and the outer submucous plexus (OSP)—near the inner side of muscular layer [30]. The ENS regulates the majority of functions of the gastrointestinal tract in physiological conditions and plays important roles in adaptive and/or regenerative processes under pathological and toxic factors [31]. The main manifestation of these adaptive processes in the ENS are changes in neurochemical characterization of the enteric neurons, i.e., changes in the synthesis and secretion of active substances acting as neurotransmitters and/or neuromodulators [30].

Given that bisphenols penetrate to the living organism mainly through the digestive system [3,4], the stomach and intestine, and therefore ENS, are highly exposed to their adverse effects. It is confirmed by previous studies, which found that even low doses of BPA administered for a relatively short period may influence the neurochemical characterization of the enteric neurons in domestic pigs [30,32,33]. The impact of bisphenols on the ENS is connected with the relatively well known ability of them to bind to estrogen receptors, which are expressed not only on the neurons located in the intestinal wall, but also in glial cells within enteric ganglia, as well as in the interstitial cells of Cajal cooperating with the enteric neurons in intestinal motility regulation [34,35]. Previous studies have also reported that activated estrogen receptors affect differentiation and functions the ENS [36] as well as inhibit the secretory activity of the stomach [37]. Moreover, it is known that that estrogen may clearly affect the nitric oxide synthase activity in the colon through estrogen receptor alpha [38] and inhibit the colonic motility [39]. Therefore, bisphenols binding to these receptors may influence on the ENS to a great extent through typical commonly known estrogenic pathways, although other as yet unrecognized pathways of bisphenol influence on the cell may exist (Figure 2). However, till now there is no information regarding the effects of BPS on the intestinal innervation, and no studies were performed concerning the effects of BPA on the ENS in mice.

Therefore, the aim of the present study was the comparison of the influence of long-term oral exposure to BPA and BPS on the neurochemical characterization of neurons within the ENS in the descending colon of mice. It should be underlined that the selection of the intestine segment is not accidental. Although the majority of ingested bisphenols are absorbed by mucosal layer of small intestine, after undergoing glucuronidation process being excreted back into the intestinal lumen as bisphenol glucuronids, they are again deconjugated by microorganism in the caecum and may act directly on the wall of the colon [40]. Therefore, the colon is exposed to the harmful effects of both bisphenols and their metabolites, because bisphenol glucuronids are also known to have estrogenic effects [41]. Moreover, the colon is the segment of the gastrointestinal tract, where many pathological processes, including cancer or inflammatory bowel disease may develop. Previous studies have reported both correlations between exposure to BPA and some of these processes [42,43] as well as the roles of the ENS in their pathogenesis [44,45]. Therefore, the knowledge on the effects of BPA and BPS on the enteric neurons in the colon may be the first step to better understanding diseases occurring in this segment of intestine, and thus probably more effective treatment of them in the future.

## 2. Materials and Methods

35 mice (CD1 strain) of both gender (20 females and 15 males) at the age of 3 months old and 30g body weight (b.w.) were included into the study. The mice were kept in the animal house of the faculty of Veterinary Medicine, University of Warmia and Mazury in Olsztyn (Poland) under the following conditions: constant temperature 22 ± 2 °C, humidity 55 ± 10%, light—dark cycle 12:12 h. The animals have access to water and food ad libitum. All experimental procedures have been performed according to the approval of the Local Ethical Committee on the Experimental Animals in Olsztyn (Decision Nº 46/2019).

At the start of the experiment mice were randomly divided into 5 groups, 7 animals in each. There were both males and females in each group. The first group—control group (C group) consists of animals without any experimental activities. The other four groups were experimental groups. The scheme of the procedure in these groups was as follows: BPAI group—animals received BPA (catalog no. 239658, Sigma Aldrich, Poznan, Poland) in a dose of 5 mg/kg b.w., BPAII group—animals were treated with BPA in a dose of 50 mg/kg b.w., BPSI group—animals were treated with BPS (catalog no. 43034, Sigma Aldrich, Poznan, Poland) in a dose of 5 mg/kg b.w. and BPSII group—mice received BPS in a dose of 50 mg/kg b.w. In all groups the administration of bisphenols was carried out in the same manner in drinking water according to the method described in previous studies [46,47].

The rationale for the doses used in this investigation is that in the light of previous studies the BPA dose of 5 mg/kg b.w. has been described as a systemic no-observed-adverse-effect level (NOAEL) dose of this substance in CD1 mouse, and the dose of 50 mg/kg b.w. is a lowest-observed-adverse-effect level (LOAEL) dose for BPA in mouse [48,49]. In order to give corrected doses of bisphenols, according to the body weight, the animals were weighted once a week.

Bisphenols were administered for three months, and after this period all animals were euthanized by decapitation. Immediately after death, the distal colons were collected and fixed in 4% paraformaldehyde (pH 7.4) for 24 h. Then, tissues were rinsed in phosphate buffer (0.1 M, pH 7.4, at 4 °C) for three days with daily buffer change. After this period colons were put into 18% phosphate-buffered sucrose solution and kept at 4 °C for at least three weeks. Then, fragments of colons were frozen (−22 °C), cut into 10-μm-thick sections with microtome (Microm, HM 525, Walldorf, Germany) perpendicular to the lumen of the intestine, and sections were mounted on the microscopic slides. Slides were stored at −22 °C until further analysis.

Fragments of tissues were subjected to routine double labelling immunofluorescence technique, according to the method described previously by Szymanska et al. [30]. This technique consisted of several successive stages (all stages were performed at room temperature -rt, stages 2–4 in humid chamber, between each stage the rinsing in PBS for 30 min. was performed): (1) drying the slides for 1 h; (2) incubation with solution consisted of 10% normal goat serum, 0.1% bovine serum albumin, 0.01% NaN_3_, 0.25% Triton X-100 and 0.05% thimerosal in PBS (for the inhibition of non-specific staining) for 1 h; (3) incubation with the mixture of two commercial primary antisera. One antibody was directed against protein gene product 9.5 (PGP 9.5—used as a panneuronal marker) and the other against one of the following neuronal factors: substance P (SP), galanin (GAL), vasoactive intestinal polypeptide (VIP), neuronal isoform of nitric oxide synthase (nNOS—a marker of nitrergic neurons) or vesicular acetylcholine transporter (VAChT a marker of cholinergic neurons) Incubation was performed overnight. Specification of antibodies is presented in Table 1; (4) incubation with species-specific secondary antibodies (Table 1) conjugated with fluorochromes to visualisation the complexes of “antigen-primary antibody”; (5) treated with buffered glycerol and cover with coverslips. To exclude non-specific labelling the typical tests were made up. These tests included pre-absorption, omission and replacement of primary antibodies by non–immune sera.

The labelling fragments of the distal colon were analysed with immunofluorescence microscope BX51, Olympus, Japan) with appropriate filters. To determine the percentage of neurons immunoreactive to particular neuronal factors at least 300 cells containing PGP-9.5 in each type of enteric plexus from each animal were analysed for the presence of each neuronal factor included in the study. The number of analysed PGP-9.5—positive cells was regarded as 100%. In order to avoid double counting of the same neurons the sections of colon included into the microscopic analysis were located at least 200 µm apart. The obtained results were pooled and presented as mean ± SEM.

To evaluate the eventual bisphenols influence on the total population of the enteric neurons, the number of all cells immunoreactive to panneuronal marker PGP 9.5 in each kind of intramural ganglia and in each animal were counted in 50 ganglia located on at least 10 slides (sections of colon were located at least 200 µm apart).

For statistical analysis the Anova test (Statistica 13, StatSoft, Inc., Cracow, Poland) was used, and the differences were considered statistically significant at *p* ≤ 0.05.

Moreover, fragments of the colon were collected for histopathological studies. After collection the fragments of the colon were put into a 10% formalin solution and then were subjected to routine hematoxylin and eosin staining at the Department of Histology, Faculty of Veterinary Medicine, University of Warmia and Mazury in Olsztyn (Poland).

## 3. Results

During the present investigation all neuronal factors studied have been observed in the enteric plexuses in the mice distal colon. All the following data showing the immunoreactivity of enteric neurons to particular neuronal factors are presented of percentage of neurons containing substances studied in relation to the number of neurons immunoreactive to panneuronal marker PGP 9.5 (treated as 100%). In the MP the largest percentage of neuronal cells contained VAChT (48.55 ± 3.66% of all cells immunoreactive to PGP 9.5). A slightly smaller percentage of the cells showed the presence of nNOS (38.83 ± 1.49%) (Figure 3), VIP (37.14 ± 1.68%) and/or SP (31.02 ± 1.49%). In turn, GAL-containing neurons were the least numerous group of cells, and their percentage amounted to 17.53 ± 1.95% of all PGP 9.5-positive cells (Figure 3).

In the SmP of the control animals the largest neuronal population constituted cells immunoreactive to VIP (28.64 ± 1.13% of all PGP 9.5-positive cells). The percentage of neurons containing VAChT and/or nNOS amounted to 21.64 ± 1.38% and 20.25 ± 0.97%, respectively. The least numerous were cells immunoreactive to SP (13.20 ± 1.05%) (Figure 3) and/or GAL (13.00 ± 0.73%).

Both bisphenols used in the investigation caused the changes in the number of neuronal cells immunoreactive to all neuronal factors studied, and intensification of changes depends on the type of bisphenol and dose of substance.

Both doses of BPA caused an increase in the percentage of neurons containing all active substances investigated. In the MP the most visible changes were noted in the case of VIP-positive neurons. Their percentage increased to 50.39 ± 1.69% of all PGP 9.5—positive cells (by about 13 percentage points—pp) under the impact of the lower dose of BPA and to 64.86 ± 1.49% (by about 27 pp) after administration of the higher dose of BPA. GAL-positive neurons underwent changes of similar intensity (Figure 3). Their percentage amounted to 29.38 ± 2.38% (increase by about 12 pp) in animals of BPAI group and to 44.18 ± 3.09% (increase by about 27 pp) in animals of BPAII group. Slightly less visible changes concerned SP-positive cells, percentage of which achieved 43.5 ± 1.83% of all PGP 9.5—positive cells (increase by about 12 pp) and 53.21 ± 4.14% (increase by about 22 pp) under the impact of lower or higher doses of BPA, respectively. In turn, the percentage of myenteric neurons containing VAChT increased to 60.12 ± 1.34% of all cells immunoreactive to PGP 9.5 (increased by about 12 pp) in animals of BPAI group and to 66.67 ± 1.07% (increased by about 18 pp) in mice of BPAII group. The least visible changes were noted in the population of neurons containing nNOS. Their percentage under the impact of lower dose of BPA increased to 48.42 ± 3.83% of all PGP 9.5-positive neurons (increase by about 10 pp) and to 52.17 ± 4.05% (increase by about 14 pp) after administration of higher dose of BPA.

In the SmP BPA also caused the increase in all populations of neurons studied, but the most visible changes were noted in the case of VAChT-positive cells. Their number increased to 41.85 ± 1.39% of all PGP 9.5—positive cells (increase by about 20 pp in comparison to control animals) in BPAI group and to 69.44 ± 1.91% (increase by about 48 pp) in BPAII group. The percentage of neurons containing VIP also underwent clear changes and amounted to 42.89 ± 2.29% (increase by about 14 pp) and to 60.78 ± 3.12% of all PGP 9.5—positive cells (increase by about 22 pp) in BPAI and BPAII groups, respectively. The populations of GAL- and/or SP-positive neurons located in the SmP underwent similar changes under the impact of BPA (Figure 3). The lower dose of this substance caused the increase in the percentage of such neurons by about 8 pp (to 21.35 ± 2.62% in the cases of neurons containing GAL and to 21.11 ± 2.75% in the case of SP-positive cells). In animals that received higher dose of BPA the number of both these populations increased by about 18 pp (to 31.53 ± 3.02% of all PGP 9.5—positive cells in the case of GAL-positive cells and to 31.40 ± 3.41% in the case of SP-positive cells). The least visible changes concerned nNOS-positive neurons. Their percentage amounted to 26.09 ± 1.48% (increase by about 6 pp) and to 36.96 ± 3.41% (increase by about 16 pp) in BPAI and BPAII groups, respectively.

BPS, similarly, to BPA, caused the increase in the percentage of neurons immunoreactive to all neuronal factors studied in both enteric plexuses. In the MP the most visible changes were noted in the case of neurons containing VIP and/or GAL (Figure 4). The percentage VIP-positive neurons amounted to 69.74 ± 3.36% of all PGP 9.5—positive cells (increase by about 32 pp in comparison to the control animals) in BPSI group and 74.01 ± 1.82% (increase by about 37 pp) in BPSII group. In the case of neurons containing GAL these values achieved 33.37 ± 1.45% (increase by about 16pp) and 49.98 ± 1.89% of all cells containing PGP 9.5 (increase by about 32 pp), respectively. Therefore, the influence of BPS on the populations of myenteric neurons immunopositive to VIP and GAL was stronger than the impact of BPA. The stronger impact of BPS was also visible in the case of nNOS- and/or VAChT-positive neuronal cells (Figure 4). The percentage of nNOS-positive neurons achieved 50.05 ± 2.9% of all PGP 9.5—positive cells (increase by 12 pp) in BPSI group and 68.98 ± 2.63% (increase by about 30 pp) in BPSII group. In turn, the percentage of neurons containing VAChT achieved 63.09 ± 1.31% (increase by about 15 pp) and 69.09 ± 1.31% of all cells containing PGP 9.5 (increase by about 21 pp) under the impact of lower or higher doses of BPS, respectively. The stronger impact of lower doses of BPS (in comparison to BPA) was also visible with a view to SP-positive cells. In BPSI group their percentage amounted to 48.01 ± 2.86% of all PGP 9.5—positive cells (the increase by about 17 pp in comparison to control animals). In turn under the impact of higher dose of BPS the percentage of SP-immunoreactive cells achieved 53.23 ± 2.99% and was similar to that noted under the impact higher dose of BPA.

In the SmP lower and higher doses of BPS acted stronger than BPA on VIP- and/or VAChT-positive cells. The percentage of neurons containing VIP amounted to 53.12 ± 5.47% of all PGP 9.5-reactive cells in BPSI group (the increase by about 25 pp in comparison to the control animals) and 55.75 ± 4.85% in BPSII group (the increase by about 27 pp). The percentage of VAChT-immunoreactive neurons achieved 43.88 ± 4.83% in BPSI group and 56.17 ± 2.90% of all PGP 9.5—positive cells in BPSII group. These values were higher than those noted in the control animals by about 22 pp and 35 pp, respectively. The treatment of BPS also increased the percentage of neurons expressing other neuronal factors studied. The population of GAL-positive cells amounted to 21.98 ±1.26% of all PGP 9.5—positive cells in BPSI group and 49.00 ± 1.95% in BPSII group (Figure 4). The first value is higher than that noted in control animals by about 8 pp, and the second is higher by as much as 36 pp. The BPS—induced changes observed in populations of SP- and/or nNOS-positive neurons in the SmP were less visible. The percentage of SP-positive cells achieved 20.52 ± 1.69% of all cells immunoreactive to PGP 9.5 in BPSI group (increase by about 7 pp in comparison to the control animals) and 31.24 ± 4.39% in BPSII group (increase by about 18 pp). BPS in lower dose increased the percentage of nNOS-immunoreactive cells by about 9 pp. (to 29.63 ± 1.68%), and in higher dose by about 14 pp (to 34.65 ± 3.50% of all PGP 9.5-positive cells).

Moreover, differences in activity between BPA and BPS in the same doses were found. BPS in low dose caused statistically significantly clearer changes in neurons containing VIP in both type of ganglia and SP in the MP, and BPS in high dose caused the clearer changes in nNOS- and/or VIP-positive neurons in the MP and GAL-positive neurons in the SmP. In turn BPA in high dose caused the statistically significantly changes in VAChT-positive neurons in the SmP. The results obtained in the present study are summarized in Table 2.

During this study it was found that both bisphenols decreased the entire population of enteric neurons. In the MP (Figure 5) of the control animals the mean number of neurons containing PGP 9.5 counted in 50 enteric ganglia amounted to 1244 ± 33.40 cells. Under the impact of lower dose of BPA their number was smaller and achieved 1186 ± 13.09, but it was not a statistically significant difference. In turn higher dose of BPA caused statistically significant decrease in number of enteric neurons to 1074 ± 22.04 cells. The influence of BPS was more visible. The number of neurons amounted to 987 ± 27.24 and 970.3 ± 5.39 under the impact of lower and higher doses of BPS, respectively. In both these cases, the differences compared to the control group were statistically significant. Moreover, changes caused by BPS in both doses were statistically significantly clearer than changes induced by BPA in the same doses.

In the SmP (Figure 6) of the control animals the mean number of neurons counted in 50 ganglia amounted to 252.4 ± 6.52 cells. BPA caused the statistically significant decrease in the number of cells to 219.9 ± 2.27 under the impact of lower dose and to 201.1 ± 2.73 under the impact of higher dose. Administration of BPS also resulted in the statistically significant decrease in the number of neuronal cells. Their numbers achieved 223.6 ± 1.41 and 224.6 ± 3.77 in animals receiving lower and higher dose of BPS, respectively. Contrary to MP the total number of enteric neurons in the SmP under the impact of BPA and BPS were not statistically significantly different. In turn the high dose of BPA caused changes statistically significantly clearer than high dose of BPS.

It should be pointed out that during the present study the differences in animal behavior and appetite between control animals and animals received bisphenols were not observed. Moreover, administration of both doses of bisphenols did not have a statistically significant effect on the weight of the animals (Table 3), did not cause any histopathological changes in the colonic wall (Figure 7) and did not change the morphology of neurons.

## 4. Discussion

The results obtained in the present investigation clearly show that both BPA and BPS have effects on the ENS of the distal colon in mice. It can be considered that this influence is significant. Such a distinct impact of bisphenols on the colonic ENS is probably connected with the metabolism of bisphenols. It is known that ingested bisphenols are absorbed by mucosal layer of the upper part of the small intestine [40]. Part of absorbed bisphenols are metabolized to glucuronide with the help of UDP-glucuronosyltransferase in the enterocytes and extracted back to the intestine lumen. The remainder of the bisphenols is transported with the blood to the liver, where it is also metabolized to glucuronide and extracted to the intestine lumen with bile. Moreover, a certain amount of bisphenols is not metabolized in the liver, but gets into all tissues of the body with the blood.

In the caecum glucurono-conjugates of bisphenols are subjected to deconjugation to free form of bisphenols by digestive bacteria [40]. Moreover, it has been shown that these free forms may be absorbed in the colon [40]. Therefore, bisphenols may act on the distal colon in two ways: directly from the lumen of the intestine and by blood supplying the intestine wall. Moreover, it is known that glucurono-conjugates of bisphenols are not completely neutral for the living organisms and may show endocrine-disrupting potential similar or even higher than BPA [41]. So, changes noted in the present study may result from both impact of free bisphenols and their glucuronited metabolites.

The present results are in agreement with previous studies, which have reported that BPA may change the ENS in various parts of the porcine digestive tract, including stomach, as well as small and large intestines [30,32,33,50,51]. However, it should be pointed out that till now the influence of BPS on the enteric innervation has not been studied.

The similarities in the influence of BPA and BPS on the ENS, noted in the present study on one side result from the similarities in the chemical structure of these compounds, and on the other side confirm that both substances studied have a similar effect on living organisms. It is in line with the latest research, which, contrary to older studies, has proven that BPS can also be considered an endocrine disruptor and affects living organisms in a similar way to BPA [24,25,26,27]. Moreover, some investigations have reported that BPS may show a stronger endocrine disrupting activity than BPA, and its influence on some organs and internal systems is more harmful [28]. The present study, in which more significant influence of BPS compared to the impact of BPA on selected populations of the enteric neurons (including VIP-positive neurons in the MP and SmP and SP-positive neurons in the MP in the case of lower doses and nNOS- and/or VIP-positive neurons in the MP and GAL-positive neurons in the MP in the case of the high dose) has been found, seems to confirm this fact. Moreover, BPS had a stronger effect than BPA on the entire population of neuronal cells in the MP.

In light of the present study, it can be concluded that bisphenols affect various populations of the enteric neurons and influence neuronal factors playing different roles in the regulation of intestinal functions. Neural factors included in this study are characterized by multidirectional impact on the intestine. Namely, acetylcholine (VAChT is a marker of cholinergic neurons)—the main stimulatory neurotransmitter in the ENS and SP are known as substances, which increase the contractility of the muscles of the intestinal wall and influence the gastrointestinal secretion [52,53,54]. SP is also an important factor, taking part in the conduction of sensory stimuli [55], and acetylcholine shows anti-inflammatory properties [56]. In turn nNOS (a marker of neurons containing nitric oxide) and VIP are one of the most important factors that inhibit smooth muscle contractility and cause relaxation of the intestine wall [57]. Moreover, both these substances are vasodilators [58,59]. The next neuronal factor included in this investigation—GAL is also involved in the intestinal motility and secretory activity, but its roles depend on the animal species and the segment of the digestive tract [60]. Moreover, the majority of the above-mentioned neuronal factors take part in modulation of the immune system [56,61,62].

Interestingly, although neuronal factors studied play such various physiological roles, the character of observed changes is similar. For example, bisphenols caused an increase in the percentage of both cholinergic neurons, which stimulate muscular contraction, and nitrergic and vipergic cells playing role in relaxation of the intestinal muscles.

Analyzing the present results concerning the percentage of cells immunoreactive to nNOS and VAChT it can be stated the colocalization of these two neuronal factors in the neurons located in the MP. Especially high degree of colocalization of nNOS and VAChT was noted under the impact of higher doses of bisphenols. Interestingly previous studies on the ENS in physiological conditions have reported that nitric oxide first of all is present in non-cholinergic and non-adrenergic inhibitory enteric motoneurons [63]. In some studies, the co-localization of nitric oxide and acetylcholine in the same neurons has not been observed at all [64]. In other investigations such co-localization has been noted, but it mainly concerned a small group of neurons—the descending interneurons in the MP [65,66]. In the light of previous studies, the degree of co-localization of acetylcholine markers (VAChT or ChAT) and nNOS clearly depends on the species and age of mammals and segment of the digestive tract studied. For example, in guinea pig small intestine such co-localization was noted in 5% on neurons [66], in human gastric fundus in about 8% of neurons [67] and in domestic pig distal large intestine—fluctuates from 30% in the mid part of gestation to 8% in adult animals [68]. Therefore, taking into account previous studies and current results, it can be concluded that bisphenols clearly increase the degree of co-localization of acetylcholine and nitric oxide in the same neurons. Due to the fact that both these factors are involved in the regulation of intestinal motility, such situation may be connected with the influence of bisphenols on smooth muscle contractility [69]. It may be connected with estrogenic activity of bisphenols, because it is known that estrogen affects nitric oxide synthase activity in the mouse colon through estrogen receptor alpha [38] and may influence on the colonic motility by the increase in nitric oxide in the myenteric neurons [39], but the exact mechanism of this phenomenon is not known and needs further studies.

Moreover, the increase in the number of neurons containing pro- and anti-inflammatory factors has been noted. The reason for such phenomenon is unknown, but a similar situation was noted in previous studies [32,33]. Probably it results from the fact that changes in neuronal immunoreactivity may be caused by various mechanisms, including fluctuations in the synthesis of neuronal factors or disturbances in their transport from body cells to nerve endings. In the light of previous observations, it is known that bisphenols have a relaxing effect on the intestine [69]. Therefore, they may cause an increase in the production of relaxant factors, including VIP or nitric oxide. Whereas the increase in production of VAChT may be probably connected with bisphenols-induced inhibition of acetylcholine release and compensatory changes in neurons to maintain homeostasis.

Due to multidirectional activity of bisphenols, the exact reasons for changes observed in the present study are difficult to explain. Apart the above-mentioned relaxant impact of bisphenols on the smooth muscles, they may be also connected with relatively well known proinflammatory activity of BPA and BPS [70,71,72]. It is more likely that the majority of neuronal factors included in this study affect the functions of leukocytes and macrophages, influence cytokines levels, or modulate homeostasis within the immune system [61,73,74,75]. On the other hand, against the thesis of a pro-inflammatory basis of the observed changes, however, is evidenced by the lack of changes typical for inflammation in the histopathological examination of the colon wall. Therefore, to evaluate the role of inflammation in changes noted in the present study further studies are needed.

Another explanation of the observed changes may be a direct neurotoxic effect of the bisphenols. Such activity of bisphenols is better known in the central nervous system, where these substances contribute to disorders in the development and functioning of synapses, fluctuations in synthesis of many neuronal factors and modulation of intracellular calcium concentration [14,17,76]. The neurotoxic basis of the observed changes is all the more likely that the present study confirmed a reduction in the total number of enteric neurons under the influence of both bisphenols studied.

It cannot be ruled out that bisphenols have a similar effect on the ENS, and changes noted in the present study are a manifestation of adaptive and/or neuroprotective processes aimed at maintaining homeostasis within the nervous structures supplying the intestine. Moreover, observed changes may arise from the influence of bisphenols on the intestinal microorganisms. This thesis is supported on the one hand by the fact that bisphenols are factors that strongly affect the bacterial flora of the digestive tract [77,78], and on the other hand, the close correlations between the gut microorganisms and the ENS are relatively well known [79].

## 5. Conclusions

The present study for the first time compares the influence of BPA and BPS on the ENS. Both these substances, even in relatively low doses, affect the neurochemical characterization of enteric neurons. Changes caused by BPA are similar to those noted under the impact of BPS, but the impact of both bisphenols on the enteric neurons clearly depend on dose of substance and type of the enteric ganglion studied. BPS more strongly than BPA reduces the total number of enteric neurons in the MP. Moreover, in the case of some neuronal populations (including VIP-positive neurons in the MP and SmP and SP-positive neurons in the MP in the case of lower doses and nNOS- and/or VIP-positive neurons in the MP and GAL-positive neurons in the MP in the case of the high dose), the influence of BPS is more pronounced than those of BPA. The exact mechanisms of changes noted in the present study are unknown, but they may be first of all connected with neurotoxic activity of bisphenols, as evidenced by the fact that both bisphenols reduced the total number of enteric neurons. Moreover, changes noted in the present study may result from proinflammatory and/or immuno-modulating activity of bisphenols, as well as from the impact on intestinal microorganisms, but these hypotheses require confirmation in further research. The obtained results strongly suggest that BPA (similarly to BPA) is not neutral for the ENS and even (concerning some neural populations) it may exert a stronger effect than BPA. However, further studies are needed to elucidate the mechanisms of bisphenols impact on the ENS.

## Figures and Tables

**Figure 1 nutrients-15-00200-f001:**
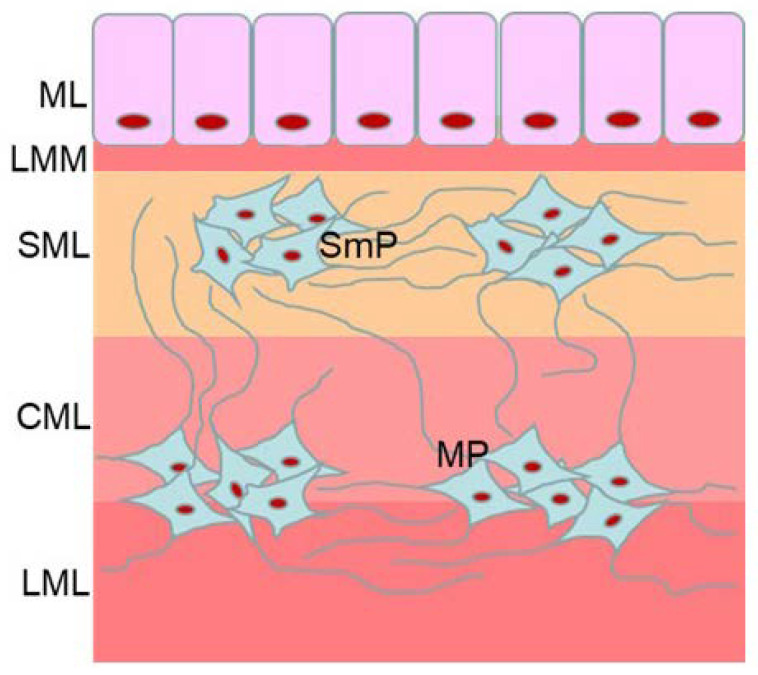
Organization of the enteric nervous system in the mouse large intestine: MP—myenteric plexus, SmP—submucous plexus, LML—longitudinal muscular layer, CML—circular muscular layer, SML—submucosal layer, LMM—lamina muscularis mucosae, ML—mucosal layer.

**Figure 2 nutrients-15-00200-f002:**
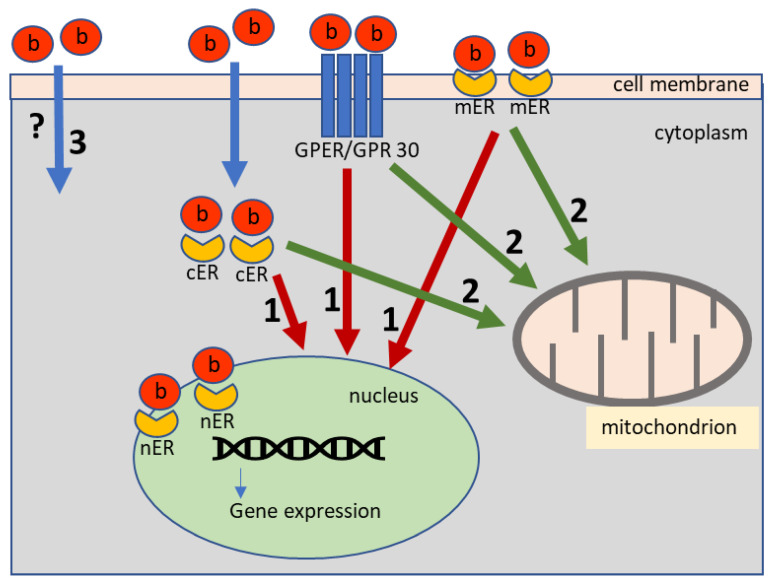
The pathways of bisphenols impact on the cell: (1) genomic estrogenic pathway; (2) non-genomic estrogenic pathway; (3) possible estrogenic independent pathway; b—bisphenols; cER—cytoplasmic estrogen receptors; mER—membrane estrogen receptors; nER—nuclear estrogen receptors; GPER/GPR 30—G protein-coupled estrogen receptor 1.

**Figure 3 nutrients-15-00200-f003:**
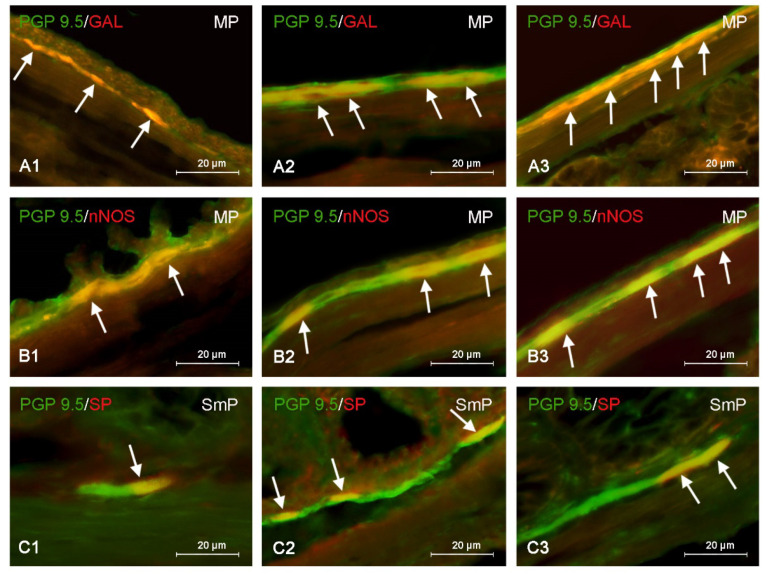
Distribution pattern of nerve cells immunoreactive to protein gene-product 9.5 (PGP 9.5)—used as pan neuronal marker and other substance studied: galanin (GAL), neuronal isoform of nitric oxide synthase (nNOS) or substance P (SP) in the myenteric plexus (MP) or submucous plexus (SmP) of mice distal colon under physiological conditions (**A1**,**B1**,**C1**) and after administration of bisphenol A in low (**A2**,**B2**,**C2**) and high dose (**A3**,**B3**,**C3**). The pictures are the result of the overlap of both stainings. The arrows are pointing neurons immunoreactive for studied substances.

**Figure 4 nutrients-15-00200-f004:**
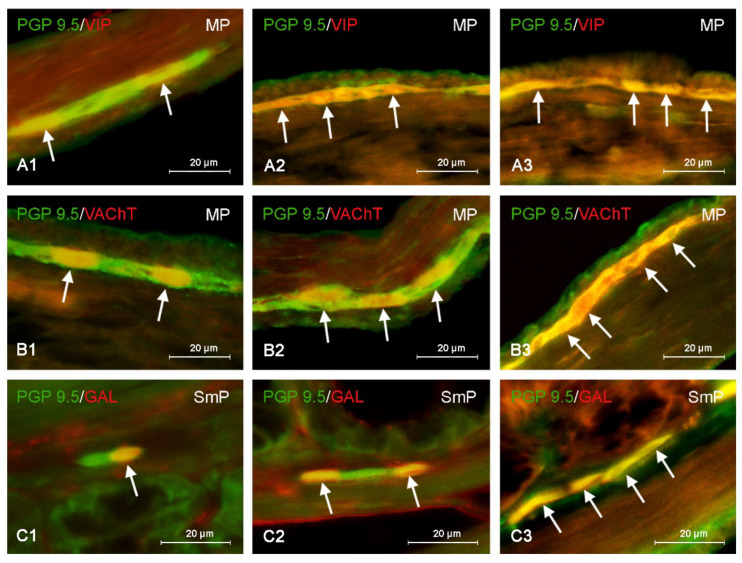
Distribution pattern of nerve cells immunoreactive to protein gene-product 9.5 (PGP 9.5)—used as pan neuronal marker and other substance studied: vasoactive intestinal polypeptide (VIP), vesicular acetylcholine transporter (VAChT) or galanin (GAL) in the myenteric plexus (MP) or submucous plexus (SmP) of mice distal colon under physiological conditions (**A1**,**B1**,**C1**) and after administration of bisphenol S in low (**A2**,**B2**,**C2**) and high dose (**A3**,**B3**,**C3**). The pictures are the result of the overlap of both stainings. The arrows are pointing neurons immunoreactive for studied substances.

**Figure 5 nutrients-15-00200-f005:**
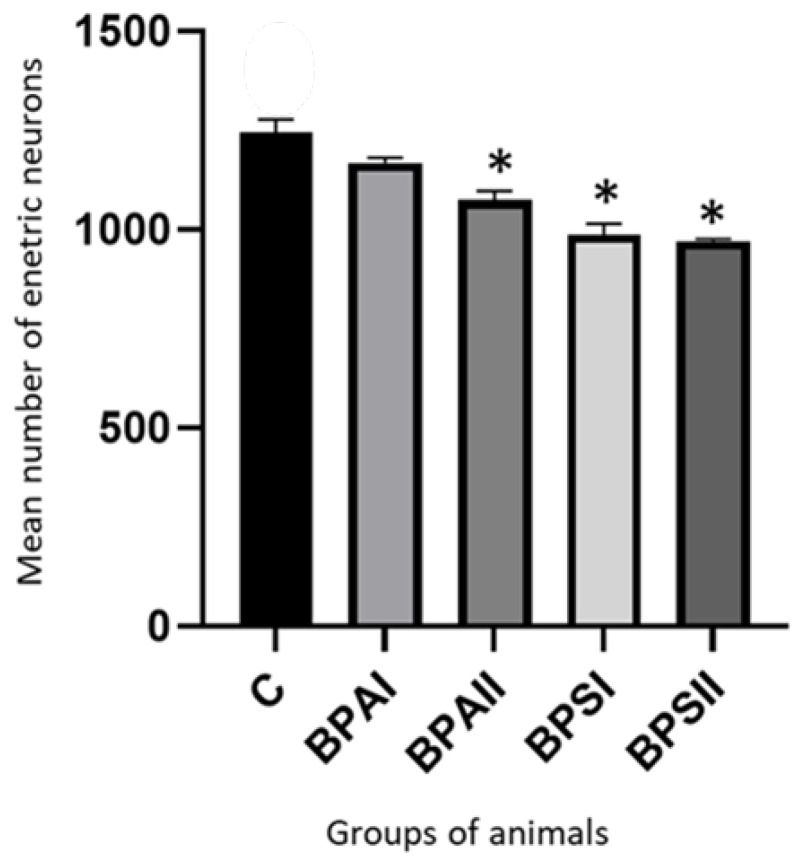
Mean number of enteric neurons (±SEM) counted in 50 MP ganglia in control animals (C), after administration of BPA in low (BPA I) and high dose (BPAII) and after administration of BPS in low (BPSI) and high (BPSII) dose. Statistically significant differences between particular groups of animals and control group are marked with *.

**Figure 6 nutrients-15-00200-f006:**
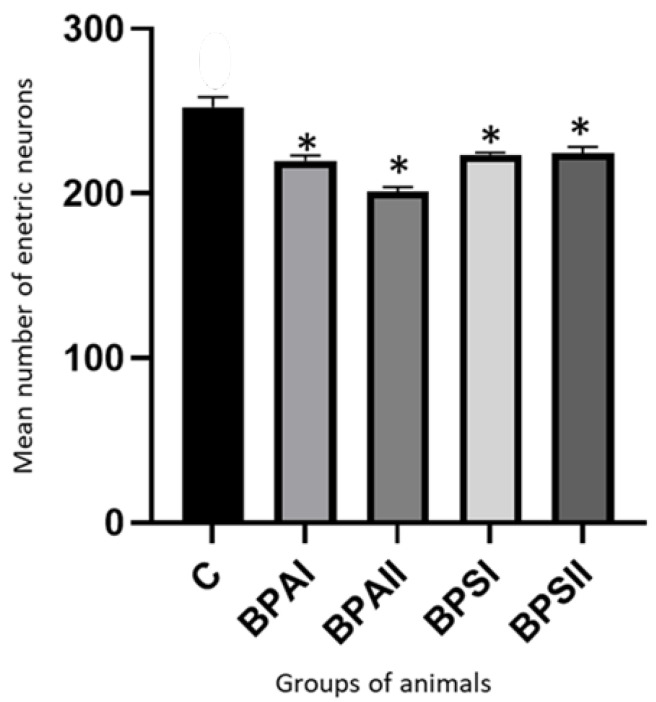
Mean number of enteric neurons (±SEM) counted in 50 SmP ganglia in control animals (C), after administration of BPA in low (BPA I) and high dose (BPAII) and after administration of BPS in low (BPSI) and high (BPSII) dose. Statistically significant differences between particular groups of animals and control group are marked with *.

**Figure 7 nutrients-15-00200-f007:**
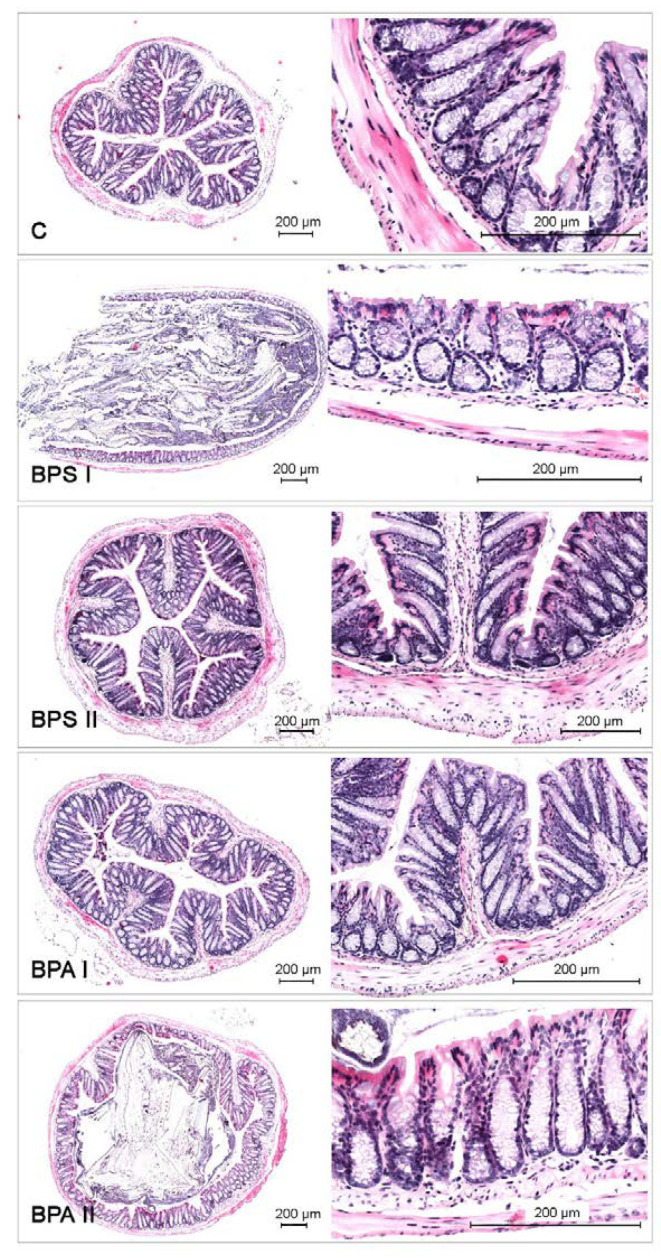
Histological staging of the colon collected from control animals (C) after administration of bisphenol S in low (BPSI) and high dose (BPSII) and after administration of bisphenol A in low (BPAI) and high dose (BPAII).

**Table 1 nutrients-15-00200-t001:** List of antisera and reagents used in immunohistochemical investigations.

Primary Antibodies				
**Antigen**	Code	Species	Working Dilution	Supplier
**PGP 9.5**	7863-2004	Mouse	1:1000	Biogenesis Ltd., Poole, UK
**SP**	8450-0505	Rat	1:1000	Bio-Rad (AbD Serotec), Kidlington, UK
**VIP**	VA 1285	Rabbit	1:1000	Enzo Life Sciences; Farmingdale, NY, USA
**GAL**	T-5036	Guinea Pig	1:1000	Peninsula
**VAChT**	H-V006	Rabbit	1:2000	Phoenix Pharmaceuticals
**nNOS**	AB5380	Rabbit	1:1000	MercMillipore, DEU
**Secondary antibodies**				
**Reagents**			Working dilution	Supplier
**Alexa fluor 488 donkey anti-mouse IgG**			1:1000	Invitrogen, Carlsbad, CA, USA
**Alexa fluor 546 donkey anti-rabbit IgG**			1:1000	Invitrogen
**Alexa fluor 546 donkey anti-rat IgG**			1:1000	Invitrogen
**Alexa fluor 546 donkey anti-guinea pig IgG**			1:1000	Invitrogen

**Table 2 nutrients-15-00200-t002:** The percentage of neurons positive for neuronal isoform of nitric oxide synthase (nNOS), galanin (GAL), vasoactive intestinal polypeptide (VIP), substance P (SP) and vesicular acetylocholine transporter (VAChT) in relation to all PGP 9.5—positive cells in the MP (myenteric plexus) and SmP (submucous plexus) in the control group (C) and all of the experimental groups (BPAI, BPAII, BPSI, BPSII).

		C	BPAI	BPAII	BPSI	BPSII
**nNOS**	MP	38.83 ± 1.49%	48.42 ± 3.83% *	52.17 ± 4.05% *^B^	50.05 ± 2.9% *	68.98 ± 2.63% *^B^
	SmP	20.25 ± 0.97%	26.09 ± 1.48% *	36.98 ± 3.41% *	29.63 ± 1.68% *	34.65 ± 3.5% *
**GAL**	MP	17.53 ± 1.95%	29.38 ± 2.83% *	44.18 ± 3.09% *	33.37 ± 1.45% *	49.98 ± 1.89% *
	SmP	13.00 ± 0.73%	21.35 ± 2.62% *	31.53 ± 3.02% *^B^	21.98 ± 1.26% *	49.00 ± 1.95% *^B^
**VIP**	MP	37.14 ± 1.68%	50.39 ± 1.69% *^A^	64.86 ± 1.49% *^B^	69.74 ± 3.36% *^A^	74.01 ± 1.82% *^B^
	SmP	28.64 ± 1.13%	42.89 ± 2.29% *^A^	60.78 ± 3.12% *	53.12 ± 5.47% *^A^	55.75 ± 4.85% *
**SP**	MP	31.02 ± 1.49%	43.5 ± 1.83% *^A^	53.21 ± 4.14% *	48.01 ± 2.86% *^A^	53.23 ± 2.99% *
	SmP	13.20 ± 1.05%	21.11 ± 2.75% *	31.40 ± 3.41% *	20.52 ± 1.69% *	31.24 ± 4.39% *
**VAChT**	MP	48.55 ± 3.66%	60.12 ± 1.34% *	66.67 ± 1.07% *	63.09 ± 2.36% *	69.09 ± 1.31% *
	SmP	21.64 ± 1.38%	41.85 ± 1.39% *	69.44 ± 1.91% *^B^	43.88 ± 4.83% *	56.17 ± 2.9% *^B^

Statistically significant (*p* ≤ 0.05) differences between C group and every experimental group are marked with *; Statistically significant differences between BPAI and BPSI group are marked with ^A^; Statistically significant differences between BPAII and BPSII groups are marked with ^B^.

**Table 3 nutrients-15-00200-t003:** Weight of animals on 1 and 91 days of experiment and weight gain observed during the investigations (g ± SEM).

	C	BPA I	BPA II	BPS I	BPS II
Animal weight on 1 day of the experiment	22.14 ± 0.74	22.71 ± 1.04	21.29 ± 1.27	22.43 ± 0.92	19.43 ± 1.46
Animal weight on 91 day of the experiment	24.14 ± 0.80	24.71 ± 1.11	23.57 ± 1.51	24.86 ± 0.83	21.86 ± 1.34
Weight gain	1.71 ± 0.29	2.00 ± 0.49	2.29 ± 0.52	2.43 ± 0.48	2.57 ± 0.48

## Data Availability

Data is contained within the article.

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
