# Peer review of "The Comparison of the Influence of Bisphenol A (BPA) and Its Analogue Bisphenol S (BPS) on the Enteric Nervous System of the Distal Colon in Mice"

_nutrients, 2022, doi:10.3390/nu15010200_

Round 1
Reviewer 1 Report
This manuscript deals with an important issue regarding safety concerns when using bisphenols in contact with food and beverages. Immunohistochemical analysis indicated increased staining for various cholinergic and non-cholinergic markers in enteric neurons. Authors concluded that bisphenols are "not neutral for enteric neurons even in low doses". This rather general conclusion should, however, be more specific. I suggest the following changes:
Major:
1) The quantitative figures are relative, not absolute numbers. This distinction is not always made throughout the text.
2) In animals treated with BPs, percentages of VAChT and nNOS positive neurons often amounted to more than 100%; e.g., in BPSII to about 140% (Table 2). This indicates that co-expression of these two markers within the same neuron which occurs normally at low levels is exaggerated by BPs. This increased co-expression is the most striking result which is, however, not properly discussed. I suggest to focus in the Discussion more on this.
3) Authors alluded to possible neurotoxic effects of BPs (Discussion, from line 327 on). However, to meaningfully deal with this issue, data on neuron loss would be required. This is, however, not possible with the sampling strategy used. To achieve this, neuron numbers in a defined area should be given.
4) In the Introduction, binding of BPs to estrogen receptors was quoted (line 40). I suggest to brouse the literature for papers on the effects of estrogen on enteric neurons. Possibly, they also increase co-expression of cholinergic and nitrergic markers?
5) Fig 1 should be corrected. There are certainly no intra-epithelial axons of SmP neurons between enterocytes. Likewise, the lamina muscularis mucosae separating the mucosa from the submucosa should be indicated.
Minor:
6) Introduction, line 94: inflammatory bowel disease comprise both, ulcerative colitis and Crohn's disease.
7) Materials and Methods, lines, 158, 159: change "fragments" for sections
Author Response
Reviewer 1
This manuscript deals with an important issue regarding safety concerns when using bisphenols in contact with food and beverages. Immunohistochemical analysis indicated increased staining for various cholinergic and non-cholinergic markers in enteric neurons. Authors concluded that bisphenols are "not neutral for enteric neurons even in low doses". This rather general conclusion should, however, be more specific. I suggest the following changes:
Major:
1) The quantitative figures are relative, not absolute numbers. This distinction is not always made throughout the text.
Answer: Indeed: values concerning neurons immunoreactive to particular substances are relative and were showed as the percentage in relation to all cells containing panneruonal marker PGP 9.5. To make it legible, information about this has been added at the beginning of the chapter (Lines 195-198). Moreover, results as well as title of table 2 have been reedited to underline relative character of these values. Moreover, absolute data concerning the influence of bisphenols on the entire population of the enteric neurons have been added (Lines 313-342).
2) In animals treated with BPs, percentages of VAChT and nNOS positive neurons often amounted to more than 100%; e.g., in BPSII to about 140% (Table 2). This indicates that co-expression of these two markers within the same neuron which occurs normally at low levels is exaggerated by BPs. This increased co-expression is the most striking result which is, however, not properly discussed. I suggest to focus in the Discussion more on this.
Answer: The authors thank the Reviewer for valuable comment. Indeed, the observation that co-localization of VAChT and nNOS increases after administration of bisphenols is interesting. According to the suggestion of the Reviewer this situation has been discussed (Lines 414-437)
3) Authors alluded to possible neurotoxic effects of BPs (Discussion, from line 327 on). However, to meaningfully deal with this issue, data on neuron loss would be required. This is, however, not possible with the sampling strategy used. To achieve this, neuron numbers in a defined area should be given.
Answer: According the suggestion of the Reviewer additional observations concerning the influence of bisphenols on the entire population of the enteric neurons have been added. (Lines 183-186, 313-342, 462-464)
4) In the Introduction, binding of BPs to estrogen receptors was quoted (line 40). I suggest to brouse the literature for papers on the effects of estrogen on enteric neurons. Possibly, they also increase co-expression of cholinergic and nitrergic markers?
Answer: Information about estrogen receptors in the ENS and their roles in the enteric neurons functions has been added to the introduction and discussion (Lines 94-103 and 432-437)
5) Fig 1 should be corrected. There are certainly no intra-epithelial axons of SmP neurons between enterocytes. Likewise, the lamina muscularis mucosae separating the mucosa from the submucosa should be indicated.
Answer: The authors thank the Reviewer for valuable comment. Figure 1 has been corrected
Minor:
6) Introduction, line 94: inflammatory bowel disease comprise both, ulcerative colitis and Crohn's disease.
Answer: Correction has been made (Line 116)
7) Materials and Methods, lines, 158, 159: change "fragments" for sections
Answer: Correction has been made (Line 180-181)

Reviewer 2 Report
1. This manuscript compared the influence of BPA and BPS on the enteric nerve system (ENS) of the distal colon in mice by double immunofluorescence technique. Both substances, even at relatively low doses, affect the neurochemical characterization of enteric neurons. In the case of some neuronal populations, the influences of BPS were more pronounced than those of BPA.
The whole subject is meaningful and worth of study. I feel that it is suitable for publication in this journal but, after the authors should accept some revisions of their paper, particularly on the following points:
2. L88-92, page3: It is indicated in the manuscript that BPA/BPS is transported back into the intestinal cavity as bisphenol glucuronides after undergoing glucoaldehyde process and then play a role. How to determine whether the distal colon of mice is affected by bisphenol or bisphenol metabolite?
3. L101, page3: In materials and methods, the sex of animals in the experiment should be supplemented.
4. L22-123, page3: In materials and methods, it was proposed in the manuscript that animals should be weighed once a week, but the results did not explain the changes in animal weight.
5. Is there any difference in animal behavior, appetite and mental state during the administration? If there is any discrepancy, it is suggested to supplement it in the results section. If there is no differences, it is recommended to explain it in the results section.
6. It is suggested to supplement pathological analysis experiment of distal colon in mice to compare the effect of BPA and BPS on the histopathology of ENC.
7. It is recommended to supplement the experimental analysis of relevant cytokine levels, such as pro-inflammatory cytokine including TNF-α、 IL-1α、IL-1β.
8. L183-185, page5: It is pointed out in the manuscript that the changes of the intensity of the two kinds of bisphenols on the studied nerve factor depend on the type and dose of bisphenols. Why only low and high doses are set for the administration mode, but not medium doses.
9. L171, page5: figure 2 needs to be renumbered, such as A1, A2, A3,B1... or Ia, Ib, Ic, IIa... etc, and shall be explained in the drawing notes.
10. L189, page5: What does “percent age points – pp” mean?
11. L189, page5: “In the MP the most visible changes were noted in the case of VIP-positive neurons Their percentage increased to 50.39±1.69% (by about 13 percent-age points – pp) under the impact of the lower dose of BPA and to 44.18±3.09% (by about 27 pp) after administration of the higher dose of BPA.” There is a error in the data in the manuscript, which should correspond to the data in table2 one by one.
12. L236, page7:figure 3 needs to be renumbered, such as A1, A2, A3,B1... or Ia, Ib, Ic, IIa... etc, and shall be explained in the drawing notes.
13. L261-264, page7: What is the meaning of “*” in column C of table2 ? Are the results obtained compared with the blank group? If yes, it is recommended to supplement.
14. The purpose of this study is to compare the effects of long-term oral administration of BPA and BPS on the neurochemical characteristics of the neurons in the colon ENS of mice. Is there a significant difference between the effects of the same dose of two different types of phenols on the neurochemical characteristics?
15. It is suggested to add some information about the mechanism of BPA affecting neuronal factors in the discussion section? What analytical methods were used? To provide ideas and reference for the study of the effect of BPS on neuronal factors.
16. L343, page9:“In the case of some neuronal populations, the influence of BPS ais more pronounced than those of BPA.” It is suggested to list specific neuronal populations in the results.
17. L343, page9: “ais” changed to “is”.

Author Response
REVIEWER 2
- This manuscript compared the influence of BPA and BPS on the enteric nerve system (ENS) of the distal colon in mice by double immunofluorescence technique. Both substances, even at relatively low doses, affect the neurochemical characterization of enteric neurons. In the case of some neuronal populations, the influences of BPS were more pronounced than those of BPA.
The whole subject is meaningful and worth of study. I feel that it is suitable for publication in this journal but, after the authors should accept some revisions of their paper, particularly on the following points:
- L88-92, page3: It is indicated in the manuscript that BPA/BPS is transported back into the intestinal cavity as bisphenol glucuronides after undergoing glucoaldehyde process and then play a role. How to determine whether the distal colon of mice is affected by bisphenol or bisphenol metabolite?
Answer: The authors thank the Reviewer for valuable comment. Indeed, determination whether observed changes are caused by bisphenols or by their metabolites was impossible during this study. On the other hand, it is known that both bisphenols and their metabolites have similar negative impact on the living organisms and the amount of bisphenols metabolites depends on the administration of bisphenols. Therefore, the negative influence of metabolites of bisphenols results from administration of bisphenols. Information about the possibility that observed changes may result from the impact of bisphenols and their metabolites has been added in the introduction and in the discussion chapters. (Lines 113-115 and 369-376)
- L101, page3: In materials and methods, the sex of animals in the experiment should be supplemented.
Answer: Sex of animals has been supplemented (lines 123 and 131)
- L22-123, page3: In materials and methods, it was proposed in the manuscript that animals should be weighed once a week, but the results did not explain the changes in animal weight.
Answer: Yes, animals were weight every week, but there were no statistically significant differences between particular groups of animals. The table with the initial and final average weight of the animals and the average overgrowth of the animals are shown in the table added into Results (Lines 345 and 346, Table 3 Lines 348-349)
- Is there any difference in animal behavior, appetite and mental state during the administration? If there is any discrepancy, it is suggested to supplement it in the results section. If there is no differences, it is recommended to explain it in the results section.
Answer: During the present study there were no visible differences in animal behavior and appetite between particular groups of animals. On the other hand, the exact analysis of mental state and cognitive functions of animals has not been performed, because this was not the purpose of the research. Information about no differences in animal behavior and appetite has been added to the results (Lines 343-345)
- It is suggested to supplement pathological analysis experiment of distal colon in mice to compare the effect of BPA and BPS on the histopathology of ENC.
Answer: Routine histopathological analysis has been made during the study. The differences between particular animal groups have not been observed. Information about histopathological studies and microphotographs have been added into the material and methods and results (Lines 189-192, Figure 6: lines 352-256)
- It is recommended to supplement the experimental analysis of relevant cytokine levels, such as pro-inflammatory cytokine including TNF-α IL-1α、IL-1β.
Answer: The authors are in agreement with the Reviewer that additional analysis would complete the manuscript and clarify additional issues related to the effects of bisphenols on the colon. Of course, the analysis of proinflammatory cytokines will allow to prove the pro-inflammatory effect of bisphenols. Unfortunately, such analysis requires additional reagents and samples, which the authors do not currently have. Therefore, analysis proposed by the Reviewer would require additional animals and agreement of ethical committee as well as implementation of the tender procedure for the purchase of reagents and thus a period of several months needed to carry out the analysis. On the other hand, pro-inflammatory activity of bisphenols are relatively well known. Previous studies have described the influence of bisphenols also on the cytokine levels in the gastrointestinal tract. Therefore, although the proposed analyzes would confirm the pro-inflammatory activities of bisphenols, they would most likely lead to results similar to those already obtained in previous studies. Moreover, the main aim of the study was to investigate the influence of bisphenols on the enteric neurons, no proinflammatory properties of these substances. On the other hand, the authors discussed proinflammatory activities of bisphenols and even concluded that the observed changes may be related to these activities. Of course, the authors are in agreement with the Reviewer that without evidence of such an effect in research, the conclusion is unfounded. Therefore, the discussion has been reworded and the pro-inflammatory properties of bisphenols were presented as one of the possible causes of the observed changes (Lines 454-457, Lines 486-488)
- L183-185, page5: It is pointed out in the manuscript that the changes of the intensity of the two kinds of bisphenols on the studied nerve factor depend on the type and dose of bisphenols. Why only low and high doses are set for the administration mode, but not medium doses.
Answer: The main aim of the study was to compare the influence of bisphenol A and bisphenol S on the enteric neurons in the colon. To doses of bisphenols, namely NOAEL as a low dose and LOAEL as a high dose have been included into the study. The first dose, which in the case of BPA did not cause any clear symptoms according to previous studies on mice was included to show whether BPS also does not cause any symptoms. It turned out that there were no symptoms, but both substances affect the neurochemical coding of neurons. The second dose was included as a dose in which some symptoms in mice was noted in previous studies to compare the influence of both bisphenols studied. Of course, other doses could be applied, but it should be taken into account that doses lower than the NOAEL would probably result in changes in ENS that are difficult to detect. In turn doses higher than the LOAEL would result in clear inflammatory changes in the intestine and in such case, it would be difficult to determine whether changes in ENS are due to the direct action of bisphenols or a secondary response to inflammation. In the case of the medium dose mentioned by the Reviewer, it should be assumed that the intensity of changes would be intermediate in relation to those observed in the experiment. The authors refrained from using this dose for ethical reasons in line with the recommendations for the protection of experimental animals, because studies would require an additional number of animals.
- L171, page5: figure 2 needs to be renumbered, such as A1, A2, A3,B1... or Ia, Ib, Ic, IIa... etc, and shall be explained in the drawing notes.
Answer: Figure 2 has been reedited according to the Reviewer’s suggestions
- L189, page5: What does “percent age points – pp” mean?
Answer: A percentage point or percent point is the unit for the arithmetic difference between two percentages. For example, moving up from 40 percent to 44 percent is an increase of 4 percentage points. This unit is also used in studies comparing the percentages in various groups of animals. The using of percentage points allows to pictorial and accurate comparison of control animals with animals receiving bisphenols and replaces the descriptive (less clear) comparison of groups of animals.
- L189, page5: “In the MP the most visible changes were noted in the case of VIP-positive neurons Their percentage increased to 50.39±1.69% (by about 13 percent-age points – pp) under the impact of the lower dose of BPA and to 44.18±3.09% (by about 27 pp) after administration of the higher dose of BPA.” There is a error in the data in the manuscript, which should correspond to the data in table2 one by one.
Answer: The authors thank the Reviewer for pointing out the error. This is an editorial error that has been corrected in the updated version of the article (Line 223)
- L236, page7:figure 3 needs to be renumbered, such as A1, A2, A3,B1... or Ia, Ib, Ic, IIa... etc, and shall be explained in the drawing notes.
Answer: Figure 2 has been reedited according to the Reviewer’s suggestions
- L261-264, page7: What is the meaning of “*” in column C of table2 ? Are the results obtained compared with the blank group? If yes, it is recommended to supplement.
Answer: The authors are in agreement with the Reviewer that * in column C was not very understandable to the reader. So, it has been removed. Symbols * indicate the statistically significant differences between particular experimental groups and control group of animals
- The purpose of this study is to compare the effects of long-term oral administration of BPA and BPS on the neurochemical characteristics of the neurons in the colon ENS of mice. Is there a significant difference between the effects of the same dose of two different types of phenols on the neurochemical characteristics?
Answer: Information about statistically significant differences between BPA and BPS in the same doses has been added into results according to the suggestion of the Reviewer (Lines 299-304, Table 2: Lines 306-310)
- It is suggested to add some information about the mechanism of BPA affecting neuronal factors in the discussion section? What analytical methods were used? To provide ideas and reference for the study of the effect of BPS on neuronal factors.
Answer: The exact mechanisms of the influence of bisphenols on the enteric neurons are not known. Probably it is connected with the relatively well-known impact of bisphenols on the estrogen receptors, which are placed on the enteric neurons, glial cells in the enteric ganglia and interstitial cells of Cajal located in muscular mucosa of the intestine and cooperating with the ENS in regulation of the intestinal motility. Information about this issue has been added into introduction and discussion according to the suggestion of the Reviewer (Lines 94-103, 432-437)
- L343, page9:“In the case of some neuronal populations, the influence of BPS ais more pronounced than those of BPA.” It is suggested to list specific neuronal populations in the results.
Answer: Conclusions has been reedited according to the suggestions of the Reviewer. Additional information has been added Lines 476-488
- L343, page9: “ais” changed to “is”.
Answer: Done (Line 483)

Reviewer 3 Report
This manuscript describes the effects of BPA on enteric neurons. The results are presented in a fluorescent form through multiple markers. However, this paper presents only a physiological description of the results, without an in-depth mechanistic study. In addition, the following adjustments need to be made.
1. In the introduction section, the authors only analyzed the toxic effects of BPA, the basic information on intestinal neurons, etc. However, the toxic effects of BPA on the broader nervous system are not presented.
2. From which signaling pathway do these alterations in neuronal markers originate? Or how does BPA affect intestinal neurons?
3, Does the morphology of intestinal cells or even intestinal neurons change after BPA treatment?
Author Response
REVIEWER 3
This manuscript describes the effects of BPA on enteric neurons. The results are presented in a fluorescent form through multiple markers. However, this paper presents only a physiological description of the results, without an in-depth mechanistic study. In addition, the following adjustments need to be made.
- In the introduction section, the authors only analyzed the toxic effects of BPA, the basic information on intestinal neurons, etc. However, the toxic effects of BPA on the broader nervous system are not presented.
Answer: The impact of bisphenol A on the nervous system has been described in the introduction according to the suggestion of the Reviewer (Lines 48-58)
- From which signaling pathway do these alterations in neuronal markers originate? Or how does BPA affect intestinal neurons?
Answer: The impact of bisphenols on the ENS is probably connected with the influence of bisphenols on estrogen receptors. Information about this issue has been added in the introduction and discussion (Lines 94-103, 432-437)
- Does the morphology of intestinal cells or even intestinal neurons change after BPA treatment?
Answer: During the present study, the influence of bisphenols on neuronal morphology has not been noted. Information about this has been added into results (Line 437)

Reviewer 4 Report
In this manuscript, Makowska and co-workers study and compare the effect of bisphenol A (BPA) and bisphenol S (BPS) on the enteric nervous System from the mouse distal colon. The authors describe the effect of these compounds based on the changes in the detection of the active substances produced by the neurons of the enteric system, by using immunofluorescence analysis of samples from mice distal colon and a number of specific antibodies.
This manuscript form part of a series of studies that started with the effect of BPA in the enteric system of the pig digestive tract, here BPS has been included in a mouse animal model. The study is interesting because it confirms a similar effect of BPS compared with BPA, and even with a more pronounced effect, on the mouse enteric system. Results are in concordance with other studies pointing to BPS as a potential hazard to human health.
Comments
Basically, the effect of BPS (and BPA) consists of an increase in the expression of neuronal factors, observed by the increasing number of positive neurons (immune-positive detected cells with the PGP 9.5 marker) for all the factors tested in this study.
Data are given as relative numbers, and one question that still remains is if the total number of neurons (PGP 9.5 positive cells in the enteric system) is maintained in BPA and BPS-treated animals, compared to controls, or whether an increase or decrease in their total number can be observed.
Non-specific immunolabeling has been excluded by using different control strategies (lines 148-150). However, a consistent PGP 9.5 labeling (green structures) is observed and even increased, in BPA-treated animals (compare controls with BPAI and BPAII in figure 2, sometimes it appears as “parallel” structures in a line or band above the labeled neurons). What do these PGP 9.5 positive structures account for? Does that labeling indicate that PGP 9.5 positive cells increase their number under certain conditions? If this is due to auto-fluorescence, why is the labeling so different among images? For instance, in the last row in figure 2, a different and more diffuse green labeling is observed in the control and BPAII conditions (also for the SP labeling in the central image, also third row). Similar structures are observed in figure 3, which are evident or not depending on the experimental condition. Please clarify what this labeling is due to, and how it was managed when the percentage of immunoreactive neurons was calculated.
Lines 275-276, in order to validate the study of the effect of BPA and BPS on the distal colon in mice, authors conclude: “Therefore bisphenols may act on the distal colon in two ways: directly from the lumen of the intestine and by blood supplying the intestine”. I would suggest the authors review the quoted reference [25] and define where bisphenols are subjected to glucuronidation, and how the glucurono-conjugates reach the intestinal lumen.
Line 189, the change observed in the VIP positive neurons at a higher dose of BPA (44.18±3.09%) does not coincide with the data shown the table 2 (64.86±1.49%).
Author Response
REVIEWER 4
In this manuscript, Makowska and co-workers study and compare the effect of bisphenol A (BPA) and bisphenol S (BPS) on the enteric nervous System from the mouse distal colon. The authors describe the effect of these compounds based on the changes in the detection of the active substances produced by the neurons of the enteric system, by using immunofluorescence analysis of samples from mice distal colon and a number of specific antibodies.
This manuscript form part of a series of studies that started with the effect of BPA in the enteric system of the pig digestive tract, here BPS has been included in a mouse animal model. The study is interesting because it confirms a similar effect of BPS compared with BPA, and even with a more pronounced effect, on the mouse enteric system. Results are in concordance with other studies pointing to BPS as a potential hazard to human health.
Comments
Basically, the effect of BPS (and BPA) consists of an increase in the expression of neuronal factors, observed by the increasing number of positive neurons (immune-positive detected cells with the PGP 9.5 marker) for all the factors tested in this study.
Data are given as relative numbers, and one question that still remains is if the total number of neurons (PGP 9.5 positive cells in the enteric system) is maintained in BPA and BPS-treated animals, compared to controls, or whether an increase or decrease in their total number can be observed.
Answer: The authors thank for valuable comment. According to the suggestion of the Reviewer additional analysis has been made. During this analysis, the influence of bisphenols on the entire population of the enteric neurons has been studied. Additional information has been added into the materials and methods, results and discussion (Lines 183-186, 313-342).
Non-specific immunolabeling has been excluded by using different control strategies (lines 148-150). However, a consistent PGP 9.5 labeling (green structures) is observed and even increased, in BPA-treated animals (compare controls with BPAI and BPAII in figure 2, sometimes it appears as “parallel” structures in a line or band above the labeled neurons). What do these PGP 9.5 positive structures account for? Does that labeling indicate that PGP 9.5 positive cells increase their number under certain conditions? If this is due to auto-fluorescence, why is the labeling so different among images? For instance, in the last row in figure 2, a different and more diffuse green labeling is observed in the control and BPAII conditions (also for the SP labeling in the central image, also third row). Similar structures are observed in figure 3, which are evident or not depending on the experimental condition. Please clarify what this labeling is due to, and how it was managed when the percentage of immunoreactive neurons was calculated.
Answer: Indeed, the microphotographs in old version of the manuscript were not of the best quality and did not clearly show the described structures. The microphotographs have been replaced with better quality photos. The labelling of green structures described by the Reviewer appeared sporadically in some sections of the colon outside the enteric ganglia in animals from all experimental groups and in control animals. They probably result from mistakes in labelling procedure (for example too much drying of the tissues). As such labelling was present in a relatively small number of slices and did not concern the enteric neurons it did not affect the counting of the cells.
Lines 275-276, in order to validate the study of the effect of BPA and BPS on the distal colon in mice, authors conclude: “Therefore bisphenols may act on the distal colon in two ways: directly from the lumen of the intestine and by blood supplying the intestine”. I would suggest the authors review the quoted reference [25] and define where bisphenols are subjected to glucuronidation, and how the glucurono-conjugates reach the intestinal lumen.
Answer: The issue related to the metabolism of bisphenols in the digestive tract has been presented in more detail in the discussion according to the suggestion of the Reviewer (Lines 362-376)
Line 189, the change observed in the VIP positive neurons at a higher dose of BPA (44.18±3.09%) does not coincide with the data shown the table 2 (64.86±1.49%).
Answer: the error has been corrected (Line 223)

Round 2
Reviewer 1 Report
All my concerns were adequately met, thank you! There are only a few minor spelling errors to be corrected (e.g., bisphenols glucuronides, change for bisphenol glucuronids)
Author Response
The authors thank the Reviewer for the positive review. Some minor spelling errors were corrected according to the Reviewer's suggestion.
Reviewer 2 Report
According to the comments of editors and reviewers, the overall quality of this manuscript has been greatly improved after the author's modification. I feel that it is suitable for publication in this journal.
Author Response
The authors thank the Reviewer for the positive review.
Reviewer 3 Report
This paper presents only a physiological description of the results, without an in-depth mechanistic study
Author Response
The authors are deeply sorry, but they do not understand the review received. The Reviewer stated that the introduction must be improved but gave no suggestions as to what issues should be added there. The authors guess that it is about mechanisms of the impact of bisphenols on the enteric neuronal cells. It should be pointed out that mechanisms of bisphenols impact on the nervous system, especially on the enteric nervous system are not fully understood. Probably they result from the binding of bisphenols to estrogen receptors and based on the relatively well-known estrogenic pathways both genomic and non-genomic. Of course other estrogen independent pathways cannot be excluded, but till now they are not known not only in the nervous system, but also in other organs Due to the fact that estrogenic pathways are relatively well described in numerous previous studies and they are commonly known, the authors think that the exact description of these pathways in the introduction is not necessary and only would make the manuscript less readable. However, to show the main pathways of the impact of bisphenols on the cell the scheme of these pathways has been added (Figure 2). Of course, the authors will be happy to improve the introduction if the Reviewer specifies what issues should be added there.
Moreover, the Reviewer stated that the mechanisms of bisphenols impact on the ENS were not studied. Indeed, the main aim of the study was to investigate the influence of BPA and BPS on the neurochemical characterization of the enteric neurons in the mice colon and this aim has been achieved. Of course, the additional analysis could provide additional information, but it is difficult for the authors to comment on the reviews, especially since the Reviewer did not indicate what additional research should be done in his opinion.
It should be pointed out the studies on the enteric nervous system presents many difficulties because analysis the level of any substances or even genes in the intestinal wall it does not answer the question whether these changes really connected with nerve cells or other structures (such as mucous membranes, glands, muscles). Therefore, such analysis could obscure the picture of the direct effect of bisphenols on the enteric nerve cells.
Practically, the only way to study the mechanisms connected with the ENS more thoroughly is through the cell culture. On the other hand, the impact of bisphenols during studies in vitro and in vivo may vary, because in vivo – a lot of factors connected with the reaction of the whole organism (for example among others the impact of intestinal microorganism or changes in the blood supply) may influence of the reaction of the ENS to administration of bisphenols. The present manuscript is an in vivo study. In the opinion of the authors the experiments in vivo, which to some extent simulate “everyday life” and results of exposure to BPA and BPS in “natural” conditions seem more justified in the initial phase of the research, and it should be emphasized that the manuscript is the first attempt in the history of research to compare the effects of BPA and BPS on the enteric nervous system. Thus, the study is a good starting point for further research on the exact elucidation of the mechanisms of action of bisphenols on intestinal neurons. Thanking for the review, the authors hope that the above explanations will allow the manuscript to be published in the Nutrients journal.
